# Dissecting the Spatially Restricted Effects of Microenvironment-Mediated Resistance on Targeted Therapy Responses

**DOI:** 10.3390/cancers16132405

**Published:** 2024-06-29

**Authors:** Tatiana Miti, Bina Desai, Daria Miroshnychenko, David Basanta, Andriy Marusyk

**Affiliations:** 1Department of Integrative Mathematical Oncology, H. Lee Moffitt Cancer Centre and Research Institute, Tampa, FL 33612, USA; tatiana.miti@moffitt.org; 2Department of Tumor Microenvironment and Metastasis, H. Lee Moffitt Cancer Centre and Research Institute, Tampa, FL 33612, USAdaria.miroshnychenko@moffitt.org (D.M.); 3Cancer Biology Ph.D. Program, University of South Florida, Tampa, FL 33620, USA

**Keywords:** tumor microenvironment, targeted therapy, therapy resistance, agent-based models, spatial statistics

## Abstract

**Simple Summary:**

Targeted therapies can induce strong tumor regression, but they typically fail to eradicate metastatic cancers. The elucidation of the causes that enable cancers to survive within a residual disease is essential for finding eradication strategies. The ability of cancers to survive eradication reflects not only cell-intrinsic sensitivities, but also microenvironmental effects. Paracrine signals produced by fibroblast, non-neoplastic cells that make tumor stroma, can provide strong but spatially limited therapy protection. Even though this phenomenon is well-established, its contribution to the ability of tumors to escape eradication remains poorly understood. To address this gap of knowledge, we developed an in silico model that recapitulates the effect of stroma on therapy responses in tumor tissues. This model enabled us to evaluate the contribution of spatial aspects of stroma-mediated resistance. Our analyses reveal that stroma dispersal might be the most important yet overlooked aspect of stromal resistance that determines the overall tumor responses to therapy.

**Abstract:**

The response of tumors to anti-cancer therapies is defined not only by cell-intrinsic therapy sensitivities but also by local interactions with the tumor microenvironment. Fibroblasts that make tumor stroma have been shown to produce paracrine factors that can strongly reduce the sensitivity of tumor cells to many types of targeted therapies. Moreover, a high stroma/tumor ratio is generally associated with poor survival and reduced therapy responses. However, in contrast to advanced knowledge of the molecular mechanisms responsible for stroma-mediated resistance, its effect on the ability of tumors to escape therapeutic eradication remains poorly understood. To a large extent, this gap of knowledge reflects the challenge of accounting for the spatial aspects of microenvironmental resistance, especially over longer time frames. To address this problem, we integrated spatial inferences of proliferation-death dynamics from an experimental animal model of targeted therapy responses with spatial mathematical modeling. With this approach, we dissected the impact of tumor/stroma distribution, magnitude and distance of stromal effects. While all of the tested parameters affected the ability of tumor cells to resist elimination, spatial patterns of stroma distribution within tumor tissue had a particularly strong impact.

## 1. Introduction

Target therapies that inhibit abnormal signaling result from strong mutational drivers, such as EGFR mutant and ALK+ non-small lung cancers (NSCLC). However, they usually fail to eradicate metastatic disease, and residual tumors eventually relapse. In some cases, the acquisition of therapy resistance might reflect the expansion of rare cells with full cell-intrinsic resistance [1,2]. However, long therapy remissions observed in many patients and direct experimental evidence [1,3,4,5,6] suggest that therapy resistance originates from partially sensitive tumor cells (persisters). Populations of therapy persisters, comprising residual disease can withstand therapeutic eradication but cannot expand under therapy. Therefore, a better understanding of the underpinning of persistence might be essential for developing therapeutic strategies to prevent therapy resistance and, ideally, to eradicate residual tumors.

The phenomenon of therapy persistence has been well documented in multiple in vitro studies as a cell-intrinsic phenomenon [1,7,8,9,10]. At the same time, the ability of tumor cells to survive therapy in vivo can reflect not only cell-intrinsic but also cell-extrinsic microenvironmental influences [11,12]. For example, co-culture with cancer-associated fibroblasts (CAFs) can strongly blunt the sensitivity of tumor cells to targeted therapies, thereby enabling the survival and proliferation of intrinsically sensitive tumor cells [13,14,15,16]. This effect of CAFs, the most abundant cellular component of tumor stroma, has been documented across a wide range of targeted therapy contexts, including inhibitors of EGFR and ALK in EGFR mutant and ALK+ NSCLC, respectively. At a molecular level, the protective effect can be mediated by multiple CAF-produced components of the extracellular matrix (ECM), such as several collagens, fibronectin and hyaluronan, growth factors (EGF and FGF family ligands, HGF, IL6, etc.), exosomal microRNA and metabolic crosstalk [12,13,14]. Mechanistically, these molecular mediators of CAF-conferred therapy protection act by activating bypass signaling (thus counteracting the effect of primary target suppression), enhancing phenotypic plasticity (facilitating adaptive rewiring of tumor cells) and increasing the threshold for apoptosis [12,13,14,17]. In vivo, these effects can lead to so-called environment-mediated drug resistance (EMDR) [11], but in contrast to the detailed knowledge of the mechanistic underpinning of EMDR, its effects within spatially heterogeneous tumor tissues are much less understood. 

Since stroma is an integral and abundant part of tumor tissues, the ability of tumors to evade therapeutic eradication might reflect the combined contribution of both cell-intrinsic persistence and EMDR. Our recent study with experimental models of targetable NSCLC indicates that, at least in some cases, EMDR has a dominant effect at the initial stages of therapy responses and the establishment of minimal residual disease [18]. For example, in the absence of therapy, the proliferation of tumor cells in the H3122 xenograft model of ALK+ NSCLC is randomly distributed through tumor parenchyma (Figure 1A). Treatment with a frontline ALK inhibitor alectinib leads to a re-distribution of patterns of cell proliferation towards peristromal spatial niches (Figure 1A), reflecting paracrine action of EMDR effects [18]. Due to the spatial limitations of the EMDR, understanding its impact on therapy sensitivity requires their consideration. The most obvious aspect is the stroma/tumor ratio and relative distribution of tumor parenchyma and stroma (Figure 1B). Additionally, the impact of EMDR depends on both the magnitude of its effect as well as the characteristic spatial distance (Figure 1B).

Despite the growing appreciation of the importance of EMDR, we are not aware of any prior studies that have attempted to dissect the contribution of distinct spatial aspects outlined in Figure 1. Addressing these questions is not feasible with standard experimental pipelines, necessitating the use of mathematical modeling tools. To evaluate the contribution of these spatial aspects, we developed a spatial agent-based model (ABM) of the EMDR effect of tumor tissue responding to targeted therapy. This ABM was initiated and parameterized based on in vivo experimental data from a xenograft model of lung cancer responses to targeted therapy. To assess the relative contribution of the distinct factors, we evaluated the impact of modulation of relevant parameter values on the residual disease size. We found that the spatial distribution of stroma has the strongest impact on the effect of EMDR. Our results underscore the importance of spatial considerations in understanding therapeutic responses in vivo. In turn, this knowledge could be used both for predictions of therapeutic responses and in the development of advanced therapeutic strategies.

## 2. Materials and Methods

### 2.1. Experimental Data

For the volumetric data and spatial analyses, we used experimental data from our recent study [18]. The volumetric data were obtained using weekly caliper-based measurements of tumor diameters. Tumor volumes were calculated assuming spherical-shaped tumors. For BrdU labeling, the animals were intraperitoneally injected with 10 mg/mL BrdU dissolved in 1X PBS, 30–45 min prior to euthanasia. Tumor tissue processing, embedding, sectioning and IHC staining were performed as described in [19]. 

### 2.2. Clinical Data

Diagnostic needle biopsy and post-surgery tissues of an ALK+ NSCLC patient were collected at Moffitt Cancer Center (Tampa, FL, USA). All tissues utilized for this study were collected as part of Moffitt Cancer Center’s Total Cancer Care protocol (MCC#14690), with patients providing written informed consent. De-identified formalin-fixed paraffin-embedded breast tissues were cut at 5-micron sections and released in support of this study under IRB-approved protocol.

### 2.3. Inferences of Proliferation and Death Rates

Changes in neoplastic population size over time, *N*(*t*) were assumed to follow the exponential growth law $N\left(t\right)=N_{0}e^{rt}$, with *N*_0_ denoting the initial population size, and *r* denoting the net growth rate. The net growth rate represents a difference between proliferation rate *p* and death rate *d*, *r = p* − *d*. Positive r results in exponential growth, while negative r results in exponential decay. Assuming that the tumor cell density does not change during the treatment, $N\left(t\right)=V(t)$, where *V* is the tumor volume, which was determined using caliper-based tumor measurement; spherical approximation was used during volume measurements. 

Proliferation and death rates were determined from the BrdU labeling index, assuming the duration of the S phase of 8 h and therefore *p* (per day) being 3-fold higher (24 h/8 h) than BrdU labeling index. Death rates were determined as the difference between net growth and proliferation rates *d = r − p.*

#### 2.3.1. Spatial Statistics Analyses

Advanced spatial statistics analyses were performed using R 4.1.2. The point pattern extraction from the Aiforia-segmented images was performed using a combination of “sf” and “Magick” packages. Spatial statistics analyses were performed using the “Spatstat”, “kSamples”, “sp” and “Goftest”) packages. Polygonal segmentation of tumor stroma was tiled/pixelated using R package “sf” and “magick” into squares with the length of an average tumor cell diameter (15 μm) [20]. X-y coordinates of the center of each square (stroma unit/pixel) were recorded and used for the subsequent analyses. The Radial Distribution function (RDF) compares the average density of points against complete spatial randomness (CSR) across different length scales [21,22]. To calculate the RDF for the distribution of marker positive/negative cells relative to the stroma, an annulus of width of 5 μm and radius *r* is placed around each of the stroma pixels. The number of marker +/− tumor cell centers within each annulus was calculated and divided by the expected number of tumor cell centers that fall inside the annulus under CSR. This calculation was repeated for each of the stromal pixels, and then the average value over all points at a certain *r* was recorded. CSR was generated in silico by maintaining the position of tumor cells and stromal pixels, as well as numbers of marker positive/negative tumor cells, but randomly shuffling tumor cell labels (marker status). For each sample, we generated a series of 39 CSF distributions and calculated a CSR average and 95% envelope of confidence.

#### 2.3.2. Agent-Based Modeling

The on-lattice ABM was developed using the Hybrid Automata Library (HAL) Java Library platform [23]. In our model, cells are set in a 100 cells × 100 cells grid; the initial distribution of tumor cells and stroma is based on the Aiforia-extracted mask of an IHC slide image representing 1500 μm × 1500 μm area. Each cell on the grid can be occupied by a tumor cell, stroma or be empty. Tumor cells are represented by agents that can divide, die, and move, while stromal pixels remain static, with a grid point reflecting an average tumor cell diameter of 15 μm. The x-y coordinated of the middle of each stroma pixel or of the nucleus of a tumor cell is assigned a point grid. Time is discrete, with each time step representing 24 h. At any timestep, the tumor cells agent can divide if there is an empty grid space in their immediate vicinity, die, or move to an available grid point in their vicinity. Stroma pixels remain fixed during the simulations. The probability of proliferation is conditioned by the proximity to the stroma, with the stroma sheltering extending to tumor cells within 3 cells from the stromal pixels. Each simulation condition result is the average of 100 simulations. The code was deposited at https://github.com/ttanya86/NSCLC_Cancers (accessed on 14 June 2024).

### 2.4. In-Silico Controls of Histological Point Patterns

A standing issue in spatial statistics is finding true random samples for a correct comparison and determining statistical significance. Given that stroma is undergoing morphological changes during treatment and thus affecting the tumor-stroma distance length ranges in the tissue samples, we needed point patterns that would capture stroma composition biases as part of true random controls. For that, we used the tissue point pattern, including the stroma and the tumor cells, but we erased the information about the proliferating status of the cells. We then randomly assigned “proliferating“ or “non-proliferating” labels to the cells, keeping either label’s initial numbers (cell densities) as in the original point pattern. We performed the spatial analysis in the same manner as for their respective tissue point patterns.

## 3. Results

### 3.1. Approach

The insufficient understanding of the spatial effects of EMDR in vivo reflects, in part, the limitations of common experimental and analytical pipelines that are primarily geared toward obtaining reductionistic molecular oncology inferences. To overcome this limitation, we sought to develop a spatial agent-based model (ABM) that captures the effects of EMDR in tumor tissues in silico. The utility of mathematical modeling-based inferences fully depends on the accuracy of the assumptions and parameter values. Thus, to ensure that the ABM is grounded in biological reality, we used our digital pathology/spatial statistics pipeline capable of quantitative spatial inferences of proliferation/death rates from histological data [19]. Using this pipeline, we extracted the inferences on proliferation dynamics as a function of stroma proximity from an experimental animal model of targeted therapy responses. These inferences were used to inform and parameterize the ABM, enabling us to systematically interrogate the effect of modulation of the different spatial aspects of stroma-mediated resistance, assessing their impact on the size of the residual tumor population (Figure 2 and Appendix A).

### 3.2. Quantitative Inferences of Experimentally Observed EMDR Effects

For the “ground truth” needed for the model parameterization, we used experimental data from a xenograft model of ALK+ NSCLC, the H3122 cell line. Upon exposure to the frontline ALK inhibitor alectinib, H3122 tumors xenografted into NSG mice demonstrate strong initial regression, followed by eventual relapse after approximately 3 months of therapy [3,18]. Even though H3122 cells are capable of surviving clinically relevant alectinib concentrations through cell-intrinsic persistence, their ability to survive therapy in vivo primarily reflects the effects of EMDR, as proliferation and survival of H3122 tumor cells within alectinib-exposed tumors are primarily limited to peristromal niches (Figure 1A). 

Tumor growth, regression and stable disease reflect the outcome of proliferation/death dynamics within the neoplastic population. When proliferation rates exceed death rates, the net growth rate is positive (tumors grow); when death exceeds proliferation, the net growth rate is negative (tumors regress). However, any given net growth rate can correspond to a very wide range of specific proliferation/death rate combinations. Differences in proliferation/death rate values can lead to substantial differences in therapy responses and evolutionary dynamics outcomes, even under identical net growth rates. Therefore, adequate recapitulation of EMDR effects requires accurate inferences on proliferation/death rates. For the subcutaneous xenograft tumors, net growth rates are easily measurable from volumetric data. Proliferation rates can be accurately assessed from BrdU labeling index. While reliable direct estimation of tumor death rates is not feasible, death rates can be accurately determined as the difference between expected (based on proliferation rates) and observed net growth rates [19]. 

To determine the net growth and decay rates of H3122 xenografts under the baseline growth and alectinib therapy, we used repeated tumor volume measurement data (Figure 3A). Under a constant tumor cell density, changes in the population size of tumor cells should be directly proportional to the changes in tumor volume. Given the global resource limitations (including space), tumor growth follows a logistic function. However, we used an exponential function, which adequately describes tumor growth within the experimental time window and does not require an estimate of the maximal population size (see Section 2). An exponential function was also used to describe tumor changes under therapy, as a negative growth rate translates into exponential decay. Using the experimental volumetric dynamics data and fitting an exponential growth law to that data (Figure 3A), we estimated the baseline growth rate of 19.2% per day and therapy-induced net decay rate of 11.3% per day. While therapy is expected to reduce tumor cell density by increasing the volume of necrotic areas and changes in tumor/stroma ratios, in the absence of more accurate readouts, volumetric data represent the best experientially available proxy of tumor cellularity. Therefore, we used volumetric net growth rates as the estimate of population growth rates.

To infer proliferation rates from the BrdU labeling index, we took advantage of a relatively invariant duration of the S phase of the cell cycle (~8–10 h) [24]. DNA replication during the S phase of the cell cycle involves the incorporation of a massive number of nucleotides; thus administration of BrdU pulse leads to a massive incorporation of the dTTP analogue BrdU [25]. Therefore, immunohistochemical staining with anti-BrdU antibody enables a binary demarcation of cells in the replication phase of the cell cycle (for an example, see Figure 1A). Assuming the 8 h duration of the S phase, a population-average proliferation rate, per day (24 h) can be found by multiplying the BrdU positivity index by 3 (24/8). Using BrdU+ labeling index inferences with our digital pathology pipeline, we found the average proliferation rates of 0.62 and 0.026 per day for the baseline and treatment, respectively (Figure 3B). Death rates were determined as the difference between the birth rate and the net growth rate. The average death rate was estimated to be 0.51 and 0.14 per day for the baseline and therapy, respectively (Figure 3C). 

Whereas tumor cell proliferation displays a seemingly random distribution at the baseline in the H3122 xenograft model, therapy induces a strong positive bias in cell proliferation to the peristromal niches with the opposite effect for the cell death due to the EMDR effects (Figure 1). To estimate this bias numerically, we used a radial distribution function (RDF) [21,26,27], comparing the observed distribution of distances of BrdU+ tumor cells to stroma to the null hypothesis of a random distribution [19] (Figure 3D). The magnitude of the RDF function maxima (g_max_) captures the magnitude of the EMDR effect, while the g_max_ location on the x axis captures the distance of the effect (Figure 3D). Our analyses revealed a flat distribution of BrdU+ at the baseline; in alectinib-treatment samples, the distribution of BrdU+ staining was strongly skewed towards the stroma edge (Figure 3E,F), with the peak corresponding to approximately 3 cell diameters away from the stroma (Figure 3E).

### 3.3. In Silico Model of EMDR

To understand the impact of EMDR on proliferation/death dynamics we developed a mathematical model of tumor tissue dynamics that explicitly considers space. To this end, we used a spatial on-lattice ABM with the Hybrid Automata Library (HAL) Java Library platform [23] (Figure 4A). The ABM captures tissue dynamics within a 100 × 100 cell grid, which corresponds to a 1500 μm × 1500 μm area of tumor cross section. In our model, the simulation time is discrete, with each step corresponding to 24 h a day. Some of the cells on the grid are occupied by stroma, which stays unchanged over the course of the simulation. The remaining cells can be occupied by tumor cells that serve as agents in the ABM. At each time step, a tumor cell can divide, die or migrate to an adjacent free space. Cell division and cell migration are contingent on the availability of a nearby free space. Therapy impacts probabilities of cell proliferation and death, with proliferation and death parameters dependent on a tumor cell location relative to a stromal edge. While therapy does not directly impact cell movement probability, increased availability of free spaces due to enhanced tumor cell elimination can enhance cell migration.

Simulations are initialized based on the spatial locations of tumor cells and stroma that were inferred from analyses of experimental samples with our digital pathology pipeline [19]. Scanned images of anti-BrdU IHC stained tumor cross sections are segmented into tumor cells (BrdU+ or BrdU−) and stroma; micronecrotic areas are excluded from analyses (Figure 4B). Continuous stromal areas are divided into pixels, corresponding to the size of a cell in the ABM grid. The segmented cross-section of a tumor is divided into 1500 μm × 1500 μm quadrants that correspond to the ABM simulation grid. 

Individual simulations are initiated with the baseline tumor cell proliferation rates. Since the ABM simulates proliferation/death dynamics in a spatially confined tissue area, we could not model an expanding population. Therefore, the death rate probability was set to match the proliferation rate. Importantly, proliferation probability is insufficient to define the proliferation rate within a simulation, as proliferation involves a space availability check (Figure 4A). A cell can be free either due to the death of an adjacent tumor cell or due to an adjacent tumor cell migrating to a nearby free cell of the grid. Therefore, parameterization involved finding a combination of proliferation, death and cell migration probabilities that provide a dynamic equilibrium with a good match with experimentally inferred proliferation rates and therapy-induced decay dynamics (Appendix A). 

Every simulation was initiated under the baseline parameters. This baseline phase served to generate, for each independent simulation run, a random initial distribution of tumor cells within a specific stromal topology, and to ensure the stability of the baseline proliferation-death equilibrium. After 350 time steps, simulation parameters were adjusted to reflect therapy initiation. To model the effect of therapy, we aimed to capture both the therapy-induced reduction in net proliferation rates, as well as the proliferation-protecting effects of EMDR. Tumor parenchyma was modeled as a binary compartment, i.e., tumor cells were either within or outside of the peristromal niche. Based on the RDF inferences, the stromal niche extended within 3 grid cells from each of the stromal cells, with the probability of cell proliferation enhanced to match the g_max_-based inferences. Cell migration probability was not impacted by the therapy, while cell death probability was set to provide a dynamic equilibrium within peristromal niches. Initiation of therapy within our in silico simulations led to a fast initial reduction in tumor population size, followed by plateauing out at the on-therapy equilibrium. The on-therapy simulations were run for 500 time steps, at which point the size of the residual neoplastic population was quantified. An example of the simulation is provided in Video S1.

### 3.4. In Silico Inferences of the Impact of EMDR

Having an in silico model calibrated to match experimentally observed effects of EMDR enabled us to start addressing the individual impact of different EMDR aspects (Figure 1B). Our simulations lacked cell-intrinsic resistance, and, in the absence of EMDR, therapy induced a complete elimination of tumor cells. Therefore, the residual population size was only dependent on the magnitude of the overall EMDR effects, enabling a “clean” in silico assessment. We started by assessing the most obvious determinants of the EMDR effects: the amount and of stroma and its spatial distribution. To this end, for the ABM initiation, we selected quadrants with different amounts of stroma: 15% for quadrant I and 6% for quadrants II–IV. Further, for the low stroma abundance quadrants (II–IV), we selected distinct topologies that differed in stroma dispersal, which affected the size of the peristromal niche (Figure 4C and Table 1). In the absence of therapy, the amount and dispersal of stroma had no significant impact on the number of tumor cells at equilibrium (Figure 5A). As expected, under a similar dispersal, a higher amount of stroma led to a larger size of residual population size (quadrants I vs. III in Figure 5A, right). However, stromal dispersal had a profoundly greater effect on the survival of tumor cells, as quadrant IV with high stroma dispersal had a significantly larger number of surviving tumor cells compared to the quadrant, despite having much lower stromal content (Figure 5A). 

Table 1 and to simulate a hypothetical EMDR-targeting therapeutic intervention, we evaluated the impact of modulating the magnitude and distance of the EMDR effects. Additionally, we evaluated the impact of modulating the proportion of stroma with the EMDR effects (Figure 4B). To this end, we addressed the impact of changing the value of the relevant ABM parameter, analyzing the resulting change in the residual population size within the size of the relevant on-treatment control (Figure 5A). For the evaluation of the proportion of EMDR-producing stroma, a subset of stromal pixels within the simulation were “inactivated”—proximity to these stromal cells had no EMDR effects (Figure 4B).

As expected, progressive reduction in all of the above parameters involved in EMDR, led to a progressive reduction in the residual population size and the eventual extinction (Figure 5B,C). Conversely, an in silico enhancement of magnitude and distance of the EMDR effect resulted in an increase in the residual tumor population size (Appendix A). The relative impact of modulating individual parameters could be assessed by the steepness of the residual population decline (decay rate). Interestingly, the relative impact of the modulation of specific parameters. Reducing the proportion of stroma with EMDR effect (EMDR producers) had the strongest impact in quadrants I and II, and IV. However, for quadrant III, reducing the magnitude and distance of the EMDR had a stronger effect on the population size (Figure 5B,C). Since the quadrants II–IV have a similar total amount of stroma, these results support the notion of the importance of considering not only the stroma/tumor ratio but also patterns of spatial stroma/tumor distribution. 

Given the strong impact of stroma dispersal and stroma abundance, we decided to further assess the interaction of these parameters by evaluating four additional quadrants with matched stroma abundance or stroma dispersal (V–VII) in Table 1, Appendix A. As expected, the largest residual population under therapy was observed in quadrant V which had the highest stromal dispersal and abundance; lower abundance and or dispersal was associated with smaller residual disease (Figure 5D). Next, we evaluated the impact of the interplay of abundance and dispersal on the effect of a hypothetical therapeutic modulation of the magnitude and distance of EMDR, as well as the fraction of activated stroma (Figure 5E,F). Quadrant V was least sensitive to the reduction in any of the aspects (reflected in the lowest decay rates), with the most pronounced loss of sensitivity to the reduction in the distance of EMDR effects (Figure 5E,F). Finally, we decided to assess the ability of the model to make predictions based on biopsy data from a primary human ALK+ NSCLC patient. To this end, we selected areas with different stromal abundance and dispersal (1–4) (Appendix A) essentially repeating the analyses shown in Figure 5. Consistent with the ABM simulations, initiated with xenograft data, higher abundance and dispersal (quadrants 3 and 4) were associated with larger residual disease size, as well as with the reduced impact of a hypothetical EMDR-targeting modulation (Appendix A).

In summary, our analyses highlight the utility of considering topological patterns of stromal dispersal on the EMDR-mediated ability of tumor cells to escape therapeutic elimination. While our analyses are insufficient to draw strong generalizable conclusions they warrant a more systematic follow-up as well as the development of predictors capable of capturing the weighted impact of stromal abundance and dispersal, perhaps with the inclusion of additional considerations.

## 4. Discussion

Despite the general recognition of the potential relevance of microenvironmental factors in therapeutic responses and advanced knowledge of the underlying molecular mechanisms, the contribution of EMDR to the ability of tumors to survive therapies in vivo remains poorly understood. To a large extent, this gap of knowledge reflects the need to consider the spatial effect, as well as the lack of relevant experimental and analytical pipelines. To start addressing this issue, we developed a spatial ABM of tumor tissue, reflecting therapeutic responses of a xenograft model of ALK+ NSCLC. This model enabled us to evaluate the relative impact of different spatial aspects of EMDR on the ability of tumors to survive therapeutic elimination. Our simulations indicate that the spatial distribution of stroma within tumors has the strongest impact on the magnitude of therapeutic responses, suggesting that it might be a highly informative prognostic factor complementary to the current molecular analyses. 

Across different types of cancers and specific therapy types, therapeutic responses ultimately boil down to the impact of therapies on altering the proliferation/death dynamics of evolving neoplastic populations by exerting cytotoxic and cytostatic effects. Tumors regress when the rates of cell death exceed the rates of cell proliferation. Proliferation and death of tumor cells, both in and out of therapy can be strongly influenced by environmental interactions, and microenvironmental heterogeneity can translate to heterogeneity of therapeutic responses within distinct microhabitats. While common cell viability assays can be used to define the impact of specific microenvironmental aspects on therapy sensitivity, their readouts represent only a snapshot that lumps cell proliferation and cell death. Recent developments in spatial molecular analysis techniques, including advanced multiplexing and even total transcriptomics analyses at the level of small regions and individual cells, enabled the analyses of actual tissues from patients or experimental animal models [28,29]. These analyses can provide a “ground truth” in terms of spatial associations between specific microenvironmental habitats and phenotypes or specific behaviors of tumor cells (such as proliferation, migration and death). These inferences could be used both for hypothesis testing and hypothesis generation. However, the histological spatial analyses are still limited to providing snapshots, which are highly useful, but insufficient for understanding the dynamic responses of neoplastic populations. Tumors represent moving targets, especially under therapeutic stress. Spatial patterns of tumor/stroma distribution typically change with treatment, as cytotoxic and cytostatic effects of therapy reduce the numbers of tumor cells, while enhanced cell death triggers wounding signals, leading to upregulated secretion of multiple growth factors and ECM components by stromal cells, in turn influencing therapy sensitivity of tumor cells and modulating immune responses. Further, the elimination of therapy-sensitive cells and selection for therapy resistance leads to changes in proliferation/rate dynamics, unfolding over space and time. 

Mathematical modeling represents the only toolset capable of making sense of the dynamic changes influenced by multiple interacting factors in space and time. Multiple prior studies have used mathematical modeling tools to dissect the contribution of tumor microenvironment to therapeutic responses [30,31,32,33,34,35,36,37]. Different aspects of therapeutic responses, including the impact of microenvironmental modifiers can be modeled using several mathematical tools, with distinct suitability towards addressing specific questions, advantages and limitations [38,39]. Two types of modeling are particularly suitable for explicit consideration of spatiotemporal dynamics: Partial Differential Equation systems (PDEs) and Agent-based Models (ABMs) built using Cellular Automata [38,39]. PDEs, such as reaction–diffusion models are capable of describing spatial gradients, and enabling capturing of spatial gradients in reaction–diffusion models [39]. ABMs treat individual cancer cells as agents with distinct properties and behaviors that can be described with probabilistic laws. Cancer cells can interact with the local microenvironment, which can be treated as static or as a different class of agents, producing emergent behaviors. Therefore, ABMs can capture different aspects of cancer, including proliferation/death dynamics, and migration, in a way that allows accounting for their change in space and time [33,40,41,42,43,44,45,46]. ABMs could be fully discrete, or combined with PDE in in hybrid ABMs, where discrete equations describe the behavior of tumor cells with continuous equations describing diffusible microenvironmental gradients [47,48]. 

Since our questions were focused on spatial proliferation-death dynamics of tumor cells, the ABM approach was uniquely suitable. Mathematical models, including ABM, could be of different complexity. In accordance with the “rubbish in rubbish out” principle, the utility of modeling output depends on the relevance of the model’s assumptions and inputs (parameters). While higher-complexity models can incorporate more biological details, they rely on more assumptions, increasing the and rise the challenge of accurate calibration [38]. The challenge is exacerbated by the insufficiency of knowledge needed to inform the model’s assumptions and parameters, as the knowledge from spectacular advances in molecular oncology studies is largely orthogonal for modeling spatiotemporal dynamics. At the same time, adequate tools to extract relevant information from experimental and clinical data are largely lacking, forcing modelers to use unvalidated assumptions and selecting parameter values from datasets of questionable relevance. 

To mitigate these limitations, we opted to use a relatively simple, fully discrete ABM approach, using a minimal set of undisputable assumptions, i.e., a cell can die, divide and move to an adjacent space, with the probabilities influenced by EMDR effects. These assumptions do not fully capture the complex eco-evolutionary dynamics during treatment adaptation. Our model does not account for the heterogeneity in proliferation, death and migration, and these parameters remain static over the course of the simulations. Our modeling simulations considered stroma to be homogeneous and static, despite the well-documented effect of therapy on activation of stromal fibroblasts [49] and stroma heterogeneity [50]. The effect of EMDR was modeled as binary (either on or off) even though the paracrine-acting component of EMDR mediators is expected to create gradients of signaling effects. The upside of these simplifications was the ability to separate the effects of EMDR without having to make a number of assumptions (rules to describe ABM behavior). Most importantly, instead of relying on merely plausible parameter values, we used quantitatively robust inferences from the recently developed spatial analysis pipeline, which enables the extraction of spatially explicit proliferation/death parameter values from experimental or clinical data [19]. In our earlier proof of principle study, we used this methodology to interrogate the microenvironmental heterogeneity in proliferation-death dynamics on therapeutic responses of TNBC to chemotherapy [19]. Here, we used this methodology to understand the impact of direct therapy protection by EMDR, dissecting the relative contribution of different spatial aspects of the EMDR effects. 

Our analyses indicate that estimating EMDR impact on therapeutic responses requires consideration of several essential factors, including stroma/tumor ratio, the spatial distribution of EMDR-producing stroma, magnitude and distance of the EMDR effects (Figure 5), as modulation of any of these parameters changes the impact of EMDR (Figure 5). These inferences have both prognostic and therapeutic implications. Our results indicate that, on their own, molecular profiling analyses (such as in [51]), or information on the stroma/tumor ratio [52,53] are insufficient to estimate the impact of EMDR on shaping tumor responses to targeted therapies. Our spatial inferences pipeline offers the tool to extract complementary data that could be used to substantially enhance existing tools. Notably, the predictive utility of stroma/tumor ratio was shown across multiple datasets, and our results indicate that the distribution of stroma could be even more important, as responses of tumors with similar stroma ratio and the same distance and magnitude of EMDR effects could vary dramatically based on the patterns of stroma (Figure 5). Similarly, our inferences could be useful for assessing the potential impact of different types of therapeutic interventions aimed at disrupting the EMDR effects. 

To the best of our knowledge, the contribution of the distinct aspects, emerging due to the spatial nature of EMDR effects (Figure 1) has not been explicitly explored in previous modeling studies. Moreover, our approach offers a unique degree of integration of mathematical modeling with experimental data. To the best of our knowledge, this degree of integration has not been achieved before for cancer modeling studies. Our study could be used as a starting point for further advances, by incorporating additional considerations (tumor heterogeneity and evolvability, stromal heterogeneity and remodeling, etc.) and increased accuracy of spatial inferences. At the same time, according to a famous aphorism, “all models are wrong, but some are useful” [54]. That is, even the most advanced and sophisticated models offer only an approximation of reality. At the same time, even relatively simple mathematical models have been indispensable for weather predictions, forestry and fishery measurements, determining dosing for irradiation therapy, etc. In this study, modeling was used to dissect the impact of EMDR on the initial therapy responses. Our inferences of the strong impact of stroma dispersal open the possibility of complementing the predictions of responses based on the stroma/tumor ratio, by developing an index that captures the spatial distribution of stroma. More importantly, our models could be a starting point towards developing the ability to predict the overall response dynamics, by incorporating considerations of cell-intrinsic resistance, therapy-induced changes in neoplastic populations and tumor microenvironment, stochastic mutational changes and other factors. 

A high stroma/tumor ratio is associated with poor prognosis in many types of cancers, including targetable NSCLC [51,55,56,57,58,59,60]. Given the well-documented evidence of stroma-mediated resistance across different therapeutic contexts [11,12,61], this correlation likely reflects EMDR effects. Our in silico evidence of a profound impact of stroma dispersal warrants a more systematic follow-up, to develop metrics capable of integrating stroma/tumor ratio with stromal dispersal and potentially, additional parameters. Since therapeutic responses reflect a combined contribution of cell-intrinsic and EMDR resistance mechanisms, spatial analyses that do not account for cell-intrinsic resistance components are not expected to be fully predictive. However, they might be useful for complementing more common expression and mutational-based analyses to predict therapy responses and guide the selection of therapeutic interventions.

## 5. Conclusions

In summary, our study used the integration of spatial data analyses of experimental tumors and ABM simulations to infer the relative contribution of different aspects of EMDR that emerge due to spatial considerations. All of the individual aspects influence the magnitude of therapy response suggesting the utility of considering for therapeutic targeting. Spatial distribution of EMDR-producing stroma might have a particularly strong influence on the outcome, suggesting the utility of incorporating the consideration of stroma dispersal into prognostic considerations, and the development of sophisticated metrics capable of weighing the incorporation of both stroma/tumor ratios and stromal dispersal patterns.

## Figures and Tables

**Figure 1 cancers-16-02405-f001:**
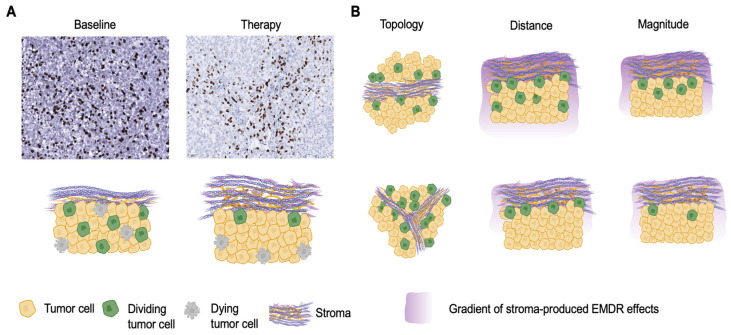
Spatial aspects of environmentally mediated drug resistance (EMDR). (**A**) Representative images of anti-BrdU immunohistochemistry staining of ALK+ xenograft tumor tissues and a schematic illustration of EMDR effects. (**B**) Schematic illustration of different aspects of EMDR that emerge due to spatial considerations.

**Figure 2 cancers-16-02405-f002:**
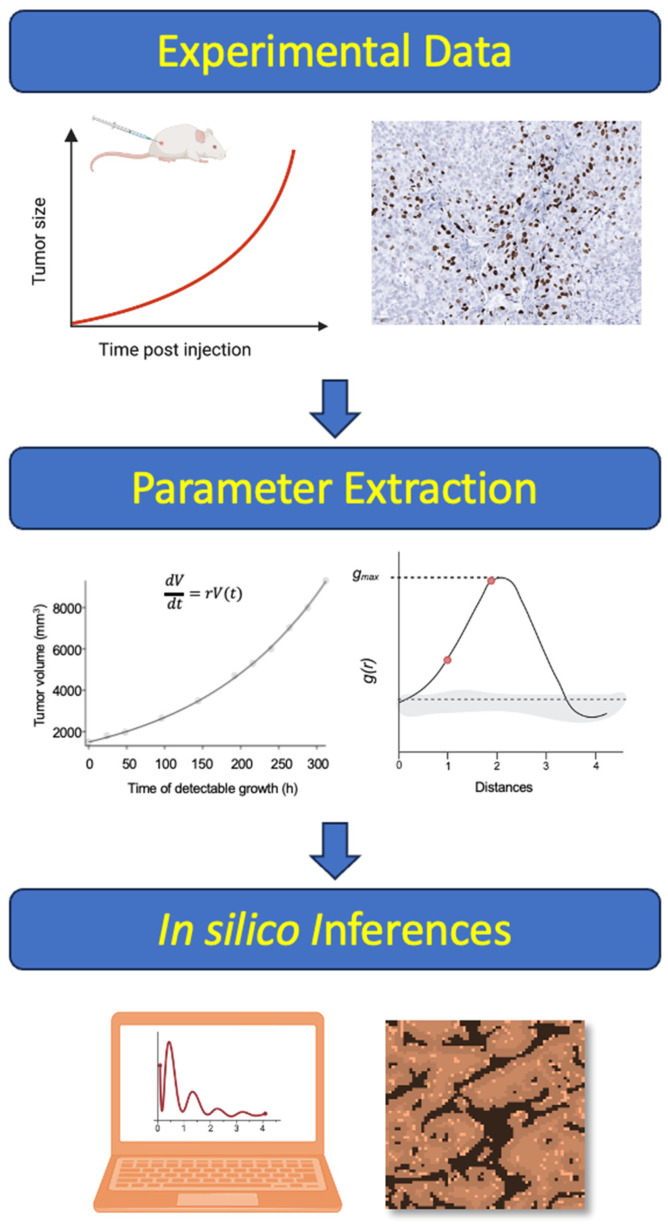
Workflow diagram of the approach.

**Figure 3 cancers-16-02405-f003:**
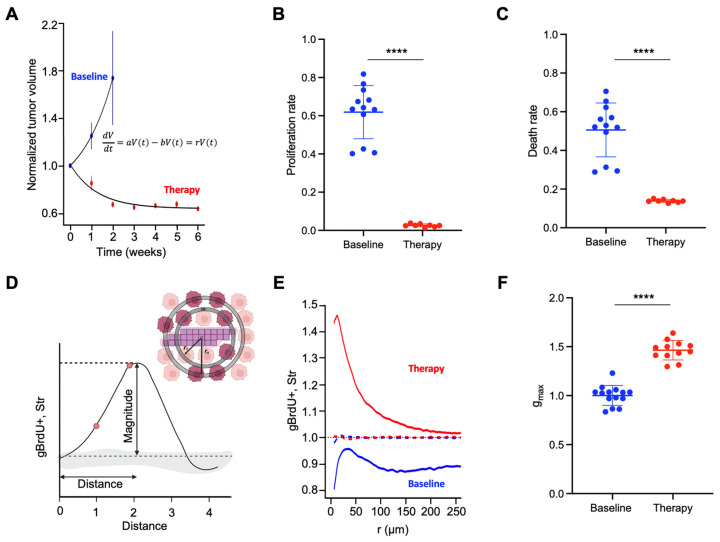
Extracting EMDR parameters from the H3122 model of alectinib responses. (**A**). Fitting the experimental tumor growth data to determine net growth rates under baseline growth and alectinib therapy. (**B**). Proliferation rate inferences, each dot represents an individual tumor. (**C**). Death rate inferences, each dot represents an individual tumor. (**D**). Schematic diagram illustrating the principle of the radial distribution function in determining the magnitude and the distance of EMDR. (**E**). An average of the RDF from individual alectinib-treated and control tumors. (**F**). G_max_ values from alectinib-treated and control tumors. **** refers to *p* < 0.0001 of Mann–Whitney statistical tests.

**Figure 4 cancers-16-02405-f004:**
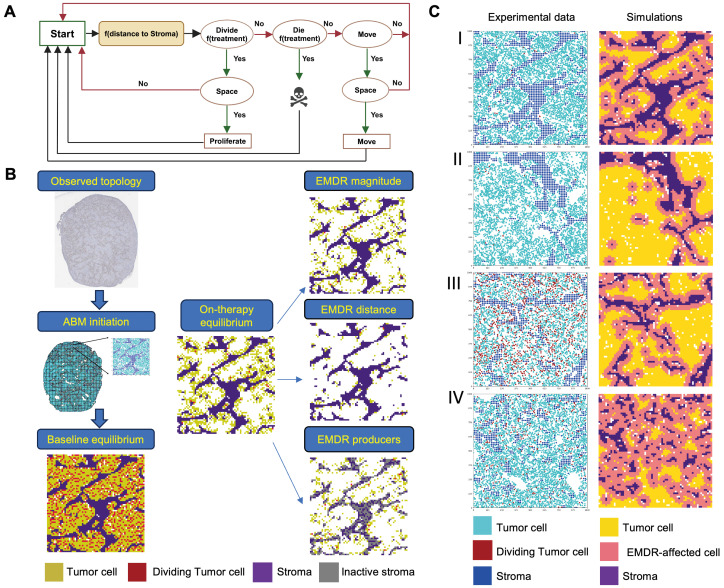
Diagram of the approach for in silico inferences of the EMDR aspects. (**A**). Diagram of the ABM. (**B**). Using ABM for evaluating the impact of different EMDR aspects within a specific stromal topology. (**C**). Evaluating EMDR effects within tissue topologies with different amounts and distributions of stroma (quadrants I–IV). EMDR parameters from the H3122 model of alectinib responses. The panel displays point patterns and snapshots of ABM simulations at the pre-treatment baselines.

**Figure 5 cancers-16-02405-f005:**
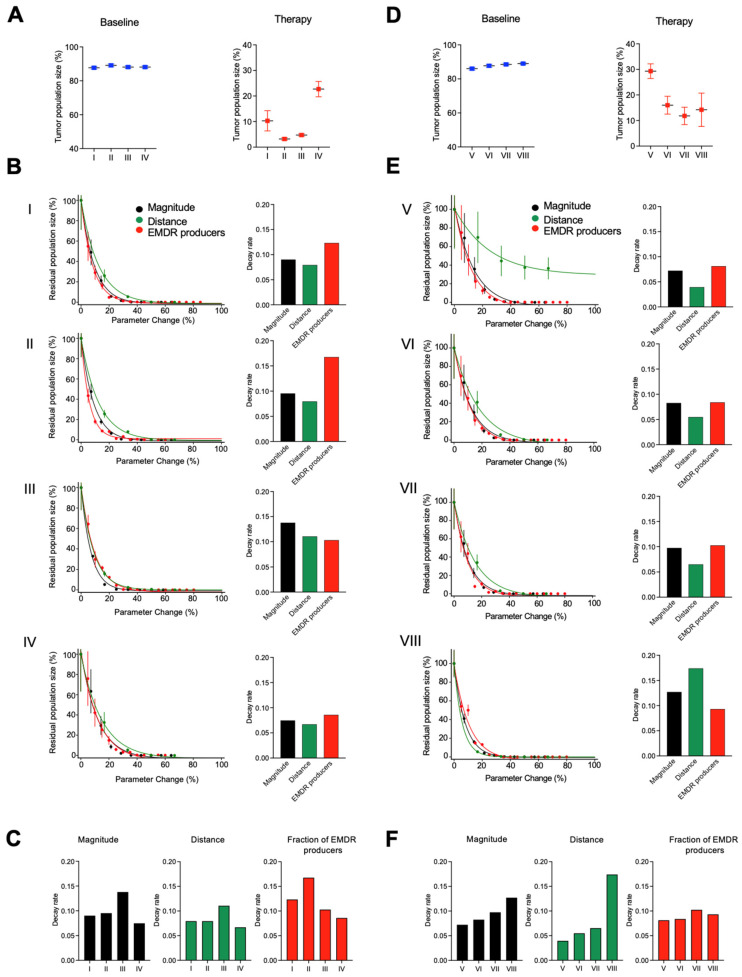
Impact of different EMDR aspects on the residual disease size. (**A**,**D**). Neoplastic population size of the indicated quadrants at a baseline and under therapy. The baseline is normalized to the total carrying capacity (maximal number of non-stromal grid cells); therapy is normalized to the average number of cells at the baseline. (**B**,**E**). Left: Effects of modulating individual EMDR aspects within different stromal topologies on the residual population size. In total, 100% correspond to pre-treatment baseline population sizes. Each dot represents the averages of 100 simulations; the line represents fits of the decay functions. The right panels display fits of the exponential decay function. Higher decay rates indicate a stronger impact. (**C**,**F**). Decay rates shown in panels (**B**,**E**) (right) are plotted for individual aspects to facilitate comparison between different topologies.

**Table 1 cancers-16-02405-t001:** Characterization of stroma abundance and dispersal within the indicated quadrants.

Tissue Quadrant	I	II	III	IV	V	VI	VII	VIII
**Stroma abundance**	15%	6%	6%	6%	22%	22%	10%	10%
**Stroma dispersal**	70%	50%	70%	90%	90%	67%	89%	67%

## Data Availability

The data from this study are available from the corresponding author upon reasonable request.

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
