# Peer review of "Dissecting the Spatially Restricted Effects of Microenvironment-Mediated Resistance on Targeted Therapy Responses"

_cancers, 2024, doi:10.3390/cancers16132405_

Round 1
Reviewer 1 Report
Comments and Suggestions for Authors
The authors constructed a spatial based model of the EMDR effect of tumor tissue in response to a targeted therapy. The authors use this to help model findings in a pre-print and ABM simulations as a first step in modeling with more realistic assumptions.
The writing was clear and understandable, but could use some rearrangements. As an example, in the results there are a lot of discussion points mixed in that would be better pulled into the actual discussion area. Also, the methods are clearly written but there needs to be access to the scripts created and/or commands used in the various steps. With the various R packages, it would also help to see the process the authors utilized for replication purposes. There are parts of the methods (definition of equations) that are within the results that should be moved. Finally, the authors did a great job listing the limitations/assumptions, but didn't compare the effectiveness of their findings compared with other similar work, this could also be extended to an alternative dataset to show reproducibility.
Some little things that need to be fixed:
Line 171: Figure 2, the I in inferences in italicized
line 304: There is a partial sentence that needs to be removed, it looks like the next sentence is what the authors wanted to say
Author Response
Responses to the Reviewer #1 comments.
We thank the reviewer for their detailed and constructive criticism. We have carefully considered all the concerns and suggestions. Below are the point-by-point responses
In the results there are a lot of discussion points mixed in that would be better pulled into the actual discussion area.”
We thank reviewer for the suggestion. While a complete separation of results and their interpretation is common in some fields, this is format is not universal and. To our opinion would be sub-optimal for the current study, as it does not represent an analysis of a new dataset using a well-established pipeline. Instead, we pose new questions and articulate the logic of our approach in addressing these questions. We believe that having a bone-try presentation of the results would make the paper less accessible to the intended broad audience. Therefore, we would prefer to keep the current format unless editorial policies dictate otherwise.
“Also, the methods are clearly written but there needs to be access to the scripts created and/or commands used in the various steps. With the various R packages, it would also help to see the process the authors utilized for replication purposes”.
We thank the reviewer for this suggestion. The scripts have been deposited as a repository in GitHub, at the following link: https://github.com/ttanya86/NSCLC_Cancers, as reflected in the revised Methods.
“There are parts of the methods (definition of equations) that are within the results that should be moved”.
Definitions of the equations have been moved to the Methods section.
“Finally, the authors did a great job listing the limitations/assumptions, but didn't compare the effectiveness of their findings compared with other similar work”
We thank reviewer for this suggestion. In the revision, we provided a brief overview of the existing literature on spatial modeling of microenvironmental mediated therapy resistance. We do not see how our findings could be compared to other work, as we are not aware of any prior study that could be used as a meaningful reference point to compare with our inferences, even not considering a specific therapeutic context (response of ALK+ NSCLC to ALKi). While mathematical oncology is a rapidly developing field, it is still small compared to the broader cancer biology/oncology field. Only a handful of studies have explicitly investigated the impact of EMDR with mathematical models, with only a subset of these considering spatial effects. Most of the prior work has focused on investigating the effect of cell-extrinsic EMDR on the evolutionary dynamics in populations consisting of therapy-sensitive and therapy resistant cells (such as Bishop at al., Nat Com 2024, cited in our manuscript). Apart from our earlier publication on the indirect chemoresistance effects of stroma (Miroshnychenko et al, Cancer Research 2023 cited in our study) we are not aware of a single study that considered proliferation/death dynamics that was based on robust inferences from directly relevant in vivo models, rather than net growth rates, or in vitro models with questionable relevance. Finally, our study aimed at dissecting the effect of EMDR, rather than fully capturing the eco-evolutionary dynamics of therapy response, as we did not consider cell-intrinsic therapy resistance, tumor heterogeneity and evolvability. Thus, we would not expect our models to accurately capture the tumor response dynamics. We reflected these points in the revised manuscript.
“this could also be extended to an alternative dataset to show reproducibility.”
We have added analyses of additional tumor topologies, both from experimental models and clinical samples. It is unclear how reproducibility could be assessed with an alternative dataset. Our results suggest that accounting for the spatial topology of tumor/stroma distribution could be more a more accurate predictor of therapy responses compared to tumor/stroma ratio. However, this work can be only viewed as a proof of principle, as using topology-based predictors would require the development and validation of such a metrics, access to relevant databases with matched pre-treatment histology and therapy outcomes, accurate segmentation of clinical samples, etc. This would require an endeavor of a substantial magnitude that is well-outside of the scope of our revised manuscript. We included the elaboration of these points in the revised discussion.
Some little things that need to be fixed:
Line 171: Figure 2, the I in inferences in italicized
line 304: There is a partial sentence that needs to be removed, it looks like the next sentence is what the authors wanted to say
We thank the reviewer for these suggestions that have been addressed in the revision.
Reviewer 2 Report
Comments and Suggestions for Authors
The authors utilized an agent-based modeling approach to analyze how various spatial aspects of tumor tissue organization influenced its resistance to treatment. Using volumetric estimates of parameter values and other assumptions, they applied the model to data images from an experimental animal model. They demonstrated that the simulations within the model resulted in the conclusion that the stroma distribution within tumor tissue had the main effect. This is an elegant study, in which the authors used a good balance between the model simplicity and available experimental information to get the dynamics of important parameters and arrive at informative insights. The manuscript is clearly written, and the conclusions are supported by the results to some extent. I have only few minor comments:
- I suppose the spatially resolved modeling should be well developed in cancer research, so more details on the current achievements and, especially, comparison of the presented results to similar studies are very desirable.
- Simulations are only based on the set of four quadrants with different stroma patterns acquired from experimental images. To enhance the conclusions, it would be nice to simulate on set-ups with more variability in the stroma patterns and other parameters.
Author Response
Responses to the Reviewer #2 comments.
We thank the reviewer for the detailed and constructive feedback. Below are the point-by-point responses to specific comments.
“This is an elegant study, in which the authors used a good balance between the model simplicity and available experimental information to get the dynamics of important parameters and arrive at informative insights. The manuscript is clearly written, and the conclusions are supported by the results to some extent. I have only few minor comments”
We thank the reviewer for their appreciation of our work.
- “I suppose the spatially resolved modeling should be well developed in cancer research, so more details on the current achievements and, especially, comparison of the presented results to similar studies are very desirable”.
Despite the rapid developments in mathematical oncology field, we are not aware no previous explicitly aimed at addressing the questions emerging from spatial nature of the EMDR that have been summarized in Figure 1. We agree with the point that spatially resolved models are reasonably well developed in cancer research. However, in our assessment the bottleneck is not on the modeling side, but rather the insufficient appreciation of the importance of spatiotemporal dynamics by the largely reductionistic experimental/clinical community with modelers. As a result, some of the important questions have not been asked, and the modeling community still suffers from the deficit of solid experimental and clinical inputs needed for development and parameterization of models. We have attempted to convey these considerations in the revised manuscript.
- Simulations are only based on the set of four quadrants with different stroma patterns acquired from experimental images. To enhance the conclusions, it would be nice to simulate on set-ups with more variability in the stroma patterns and other parameters.
We thank the reviewer for this suggestion. We have included simulations of additional quadrants that aimed to address the question of the interplay between stromal abundance and dispersal. These new analyses enabled additional useful inferences. The new data is included in the revised Figure 5 and Table 1, as well as the new Supplementary Figure 4. In addition, we also included the simulations with histological data from primary human ALK+ NSCLC tumors. This data is included in the new Supplementary Figure 5.
Reviewer 3 Report
Comments and Suggestions for Authors
The authors study the local effects of the tumor microenvironment on tumor response to targeted therapies. Their goal is to understand how microenvironmental factors, particularly paracrine signals from fibroblasts in the tumor stroma, contribute to therapy resistance. The paper characterizes tumor dynamics and its spatial distribution by weekly caliper-based measurements of tumor diameters from a xenograft model of ALK+ NSCLC using the H3122 cell line. These experimental results were used to propose and calibrate the parameters of a agent-based model that simulates tumor dynamics in a two-dimensional grid. Figure 4 illustrates the striking qualitative agreement of the ABM with the experimental data, while Fig. S2 shows that the quantitative agreement is also exceptional. Since therapy resistant cells are mainly found near the stroma, the authors conclude that the spatial pattern of stroma distribution within the tumor tissue is a key factor influencing tumor response to therapy.
The paper is very well written and the research is solid. It is certainly an important contribution to the Special Issue "Adaptive Resistance to Targeted Cancer Therapies and Rational Development of Combination Therapies". I would only suggest that the authors mention other mathematical approaches, besides ABM, that consider a tumor and its microenvironment (see, e.g., Kuznetsov et al., 2021).
line 113: delete the right parenthesis
Kuznetsov M, Clairambault J, Volpert V, Improving cancer treatments via dynamical biophysical models. Phys. Life Rev. 39 (2021) 1-48
Author Response
We thank the reviewer for appreciating the importance and quality of our interdisciplinary research. We have added a brief overview of other modeling approaches, including the suggested citation of a high-quality review and other relevant citations. This enabled us to better explain the reasoning behind choosing ABM, and to highlight the novel contributions of this study.